# Effects of Upper Body Exercise Training on Aerobic Fitness and Performance in Healthy People: A Systematic Review

**DOI:** 10.3390/biology12030355

**Published:** 2023-02-23

**Authors:** Natalie Marterer, Hendrik Mugele, Sarah K. Schäfer, Martin Faulhaber

**Affiliations:** 1Institut für Sportwissenschaft, Universität Innsbruck, Fürstenweg 185, 6020 Innsbruck, Austria; 2Leibniz Institute for Resilience Research, 55122 Mainz, Germany

**Keywords:** upper body endurance training, arm crank ergometry, wheelchair exercise, handcycle, VO_2peak_, transfer effects

## Abstract

**Simple Summary:**

Upper body endurance training is a widely underrated workout for improving endurance performance in healthy persons and is heavily understudied to date. Thus, the aim of this systematic review was to summarize the research in this field with respect to improvements in cardio-respiratory fitness. On average, upper body endurance training improved oxygen uptake and performance in the trained muscles. Evidence of transfer effects from trained arm to untrained leg muscles was inconclusive. Recommendations for upper body endurance training based on the outcome of this review include the following: (a) training programs should be performed for more than five weeks; (b) intensities greater than 70% of peak oxygen uptake in the arms should be used; and (c) continuous or interval training modes are both equally effective.

**Abstract:**

**Purpose:** This systematic review aimed to evaluate the effects of upper body endurance training (UBET) on oxygen uptake (VO_2_) in healthy persons and derive evidence-based recommendations to improve upper body fitness and performance. **Methods**: Databases were systematically searched in accordance with PRISMA guidelines until 1 February 2023. Eligibility criteria included healthy male and female adults and older adults who underwent an UBET intervention. Outcomes of interest included physical fitness (VO_2peak_ and/ or VO_2 submax_) and transfer effects (i.e., effects from trained (VO_2peak ARM_) to untrained (VO_2peak LEG_) musculature). **Results**: The search identified 8293 records, out of which 27 studies reporting on 29 interventions met our eligibility criteria. The average duration of interventions was 6.8 ± 2.6 weeks with 3.2 ± 0.8 training sessions per week. For 21 of 29 interventions, significant increases in VO_2peak ARM_ were reported following UBET (+16.4% ± 8.3%). Three of the nine studies that analyzed transfer effects of untrained legs after upper body training exhibited significant increases in VO_2peak LEG_ (+9.3% ± 2.6%). **Conclusions**: This review showed that UBET is a beneficial and useful training modality to increase the oxygen utilization in the upper body. Although UBET is an uncommon form of endurance training in healthy individuals, transfer effects to the untrained muscles can be observed in isolated cases only, rendering transfer effects in UBET inconclusive. Further research should focus on the peripheral changes in muscle morphology of the trained muscles and central changes in cardiovascular function as well as when transfer effects can occur after UBET in healthy people.

## 1. Introduction

Constant-intensity and interval endurance training are widely used to improve several health and performance indicators. Generally, the training mode consists of running or cycling, i.e., most of the current research analyzes the effects on changes in peak oxygen uptake (VO_2peak_) after exercising with large muscle groups [1,2,3], i.e., lower body endurance training. There is an increasingly large body of research for upper body endurance training (UBET) or acute effects of upper body training, i.e., exercising with small muscle groups, in rehabilitation or performance settings (e.g., professional para-handcycling) for spinal cord-injured individuals (SCII) [4,5,6]. This article is the first systematic review to summarize the effects of UBET in healthy individuals and not in SCII. It is, thereby, an important contribution to exercise science because (a) UBET provides additional opportunities for healthy adults to reach fitness goals and health benefits and (b) UBET can be used to bridge fitness during injury periods. With targeted training guidelines, this systematic review provides an evidence-based approach to UBET.

### 1.1. Physiology of UBET in Healthy Persons

It is necessary to verify whether UBET holds advantages in non-clinical populations. For evidence-based recommendations by which to improve upper body performance and to know for which individuals hand cranking is useful, central and peripheral hemodynamics in exercising humans must be considered. 

#### Comparing Upper and Lower Body Endurance Training

In a study by Calbet [7], systemic vascular conductance, peak cardiac output and stroke volume were lower during arm cranking than leg pedaling. However, mean blood pressure, the rate-pressure product and the associated myocardial oxygen demand were higher during maximal arm cranking than leg pedaling [7]. Accordingly, the estimated work performed by the heart at maximal intensity is higher than that during leg pedaling, although cardiac output was lower during arm cranking. After low-intensity training, mean O_2_ transit time in the arm is unchanged but O_2_ diffusing capacity is elevated which most likely results from increased perfusion pressure rather than enhanced vasodilation [7,8]. Elevated perfusion pressure is likely due to a higher mean arterial blood pressure during arm cranking compared to leg pedaling [9,10,11]. During combined arm and leg exercise in an upright position, the carotid baroreflex (CBR) regulates the blood pressure in direct relation to work intensity and engaged muscle mass. Therefore, the blood pressure is lower during combined arm and leg exercise than during arm cranking alone as the central blood volume is enhanced by the muscle pump of the legs and CBR resets blood pressure to a lower level [12]. The combined effect of a reduction in mean arterial pressure (MAP) and sympathetic vasoconstriction alters the distribution of cardiac output [9,10,11,13]. These earlier findings implicate that whole-body exercise is advantageous to distribute workload between arms and legs and leads to a lower amount of work for the heart compared to exercising the arms alone. This may also explain the incidence of heart attacks during gardening or snow shoveling, where upper body work and its high isometric work component are involved [14]. 

### 1.2. Transfer Effects

The knowledge of “transfer effects” defined as adaptation from trained to untrained musculature is of great interest in exercise science. Even now, it remains uncertain what training effects are transferable (i.e., increase in performance measured during exercise to untrained muscles). In a seminal study by Saltin et al., one of the earliest to assess transfer effects, peripheral (changes in the trained skeletal muscles) and central changes (changes in central circulatory functions) to one-legged exercise after 4 weeks of training were investigated [15]. One of the major findings was that all subjects demonstrated an improvement in the two-legged VO_2peak_ after one-legged training. However, improvements in VO_2peak_ and lowered heart rates and blood lactate responses at submaximal work levels were only found for the trained leg. Therefore, one-legged training likely caused some transferable improvement of the central cardiovascular function. Another transferable enhancement of oxygen uptake in non-trained arms after leg training was seen in the study by Clausen et al. [16]. The studies by Klausen et al. [17] and McKenzie et al. [18] showed that the lactate concentrations in arterial blood were significantly decreased after exercising with the trained muscle groups and did not change significantly during exercise with non-trained muscle groups. Furthermore, our own study on transfer effects found an increased cardiovascular performance measured during exercise involving untrained arms in healthy subjects [19]. In the study by Swensen et al. [20], only two-legged training induced significant gains in arm aerobic capacity whereas two-legged and one-legged training resulted in significant two-legged aerobic capacity [20]. The authors suggested that improvements in untrained muscles were dependent on the quantity of muscle mass involved in the training [20]. This raises an important question: whether effects from smaller muscle groups such as arms transfer to larger muscle groups such as legs. Physiologically, aerobic training induces peripheral adaptations (increased capillary density [21], oxidative enzyme activity [22], greater mitochondrial density [23]) in the trained muscles and central adaptations (improvement in oxygen transport and oxygen uptake by the trained and untrained muscles [16,24]), which potentially mediate transfer effects to improve exercise performance. However, studies elucidating these mechanisms, especially in the case of UBET, are rare. 

Thus, the aim of this review is to summarize research on UBET in healthy participants with respect to improvements in cardio-respiratory fitness. The findings of this systematic review could provide a foundation for evidence-based recommendations on training design, length, intensity and duration best suited to improve upper body performance assessed by physical capacity (VO_2peak ARM,_ PO_peak ARM_). In addition, submaximal parameters (VO_2submax ARM_, PO_submax ARM_) and transfer effects from arms to legs (VO_2peak LEG,_ VO_2submax LEG_, PO_peak LEG,_ PO_submax LEG_) are assessed to elucidate how UBET might be beneficial to submaximal performance and non-trained musculature fitness. 

## 2. Materials and Methods

A systematic literature search was conducted in accordance with the Preferred Reporting Items for Systematic Reviews and Meta-Analyses (PRISMA) standard [25] (Figure 1). The electronic databases of PubMed, Web of Science, Scopus and EMBASE were systematically searched until 1st of February 2023, using identical search strings (Appendix A). English- and German-language publications in non-clinical human populations with no restrictions to the study design were included. To minimize selection bias, two authors (N.M. and H.M.) independently performed the literature screening process using the software tool rayyan.ai. The search process included the removal of duplicates, the screening of titles and abstracts, as well as the assessment of full texts for eligibility. Additionally, reference lists of all potentially eligible full texts and upper body exercise-related review articles were manually checked for further eligible studies.

### 2.1. Eligibility Criteria

Eligibility was determined at the levels of title of the article and of the abstracts and full texts. Inclusion criteria were healthy adults, children and elderly persons (independent of their habitual training status) of all genders who were undergoing an upper body endurance training intervention. Exclusion criteria were patients with chronic or acute diseases (e.g., diabetes mellitus, cancer, COPD, obesity) or injuries (e.g., palsies, spinal cord injuries, traumatic injuries). All types of endurance training on an arm crank ergometer (A), a wheelchair ergometer (W) or on a handcycle (H) were eligible. Another inclusion criterion was described as at least one of the experimental groups performed upper body endurance training. Studies that examined training that involved legs (such as combined training intervention), additional upper body resistance training, functional electrical muscle stimulation, whole-body vibration or conducted training in hypoxic conditions or used special diets during the training intervention and/or tests were excluded from further analyses after the initial search.

Finally, to ensure that the effects of different UBET on oxygen uptake (VO_2_) in healthy people were evaluated, it was necessary that all final selected studies measured VO_2peak_ and/ or VO_2submax_ at pre- and post-training interventions. Furthermore, only training interventions with a minimum duration of 2 weeks were included.

### 2.2. Data Extraction

The following data were extracted from each eligible full text: (a) general study information (first author’s last name, publication year), (b) subject information (sample size, gender, age, height, weight, training status), and (c) intervention data of upper body training (training mode: length, intensity, duration, test device; test protocols). Furthermore, objective measures of physical capacity outcomes (VO_2peak ARM_, PO_peak ARM_) and VO_2submax ARM_ were assessed. Other outcomes, for example, peak and submaximal HR, LA, V_E_, RER and RPE, were only mentioned. Transfer effects, i.e., effects from arm training to changes in performance tests on a bicycle ergometer (cycling) or on a treadmill (running) (VO_2peak LEG_, PO_peak LEG_) were also extracted.

**Figure 1 biology-12-00355-f001:**
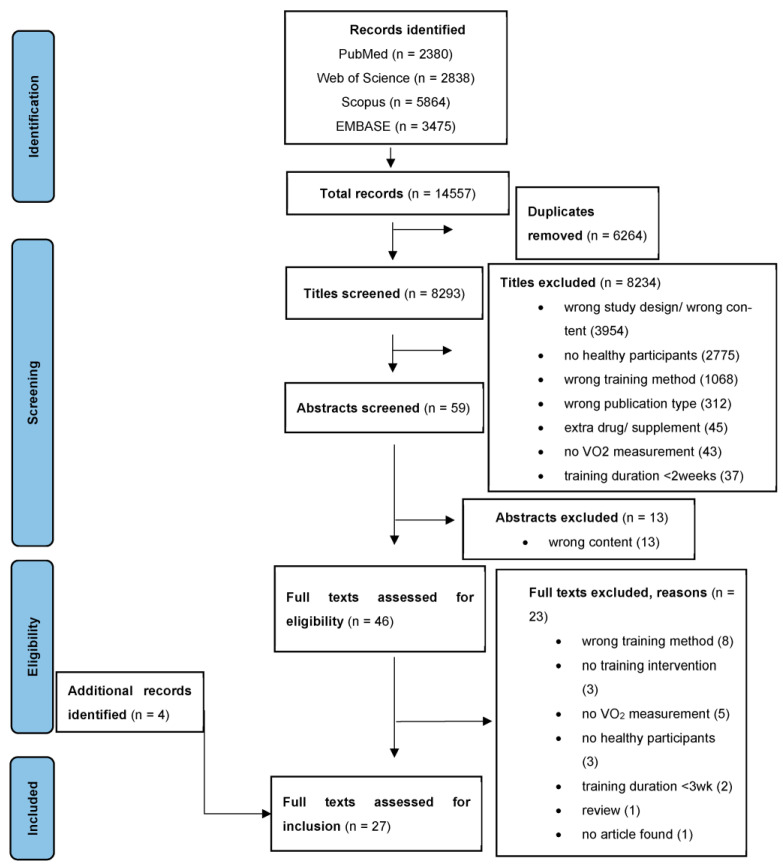
PRISMA flowchart of the systematic review process. PRISMA, Preferred Reporting Items for Systematic Reviews and Meta-Analyses.

## 3. Results

A detailed overview of individual results across the included studies is provided in Table 1, Table 2 and Table 3. A total of 14,557 studies were identified through the initial search strategy (Figure 1). In a first step, 6264 duplicates were removed. After the screening of titles and abstracts, 8247 articles were found to be ineligible and were excluded. A total of 46 full-text articles remained for further eligibility assessment. Additionally, screening reference lists of related articles retrieved a further four studies. Out of these 50 fully screened papers, 23 papers were excluded based on reasons specified in Figure 1. Consequently, 27 articles [16,17,18,24,26,27,28,29,30,31,32,33,34,35,36,37,38,39,40,41,42,43,44,45,46,47,48] were included for final evaluation. 

It must also be mentioned that the studies of Klausen et al. (1974) and Clausen et al. (1973) [16,17] and the studies of de Groot et al. (2008 and 2013) [38,39] all used the same sample for their analyses (this is indicated by grey shades in all Tables). The results were only considered once in the calculations of mean values.

### 3.1. Subject Characteristics

Table 1, Table 2 and Table 3 summarizes all 27 included studies separated by their training device. The number of subjects per study ranged between 5 and 20 with a mean value of almost 9 subjects per study. A total number of 294 participants (mean age 26 years) were included in this systematic review. More precisely, n = 157 (mean age: 29 years), n = 101 (mean age: 23 years) and n = 36 (mean age: 22 years) underwent upper body endurance training on an arm crank ergometer, on a wheelchair or on a handcycle, respectively. All subjects were healthy and able-bodied. Their training status differed from a sedentary untrained lifestyle or not actively engaged in sports over the last year [18,27,33,35,36,37,44,45] to a physically active lifestyle [16,24,26,28,29,32,39,41,42,43,46,47,48]. The training status was not described in four studies [30,31,34,40]. However, none of the participants had any specific arm training before the intervention started. 

### 3.2. Training Design

The duration of the interventions ranged from a minimum of 14 [29] days to a maximum of 12 weeks [27,33] (average: 7 weeks). The average number of weekly training sessions was three. The duration of the training session varied from 4 [41] to 70 min [38]; in most of the studies, the session lasted 30 min. As shown in Table 2, Table 3 and Table 4, 16 studies used arm cranking [16,17,18,24,26,27,28,29,30,31,32,33,34,35,36,37], 8 studies used wheelchair exercise [38,39,40,41,42,43,44,45] and 3 studies used handcycling [46,47,48] as the training mode for upper body exercise. There was no resistance training or strength training incorporated into these studies. The training intensities in the studies varied markedly, using different indicators for workload and ranging from 30 to 90% of the heart rate reserve (HRR), peak heart rate (HR), VO_2peak_ or PO_peak_. Thirteen studies investigated a continuous training program [24,26,27,28,33,36,37,38,39,44,45,46,47], twelve studies investigated an interval training program [16,17,18,29,30,31,34,35,40,41,42,43] and two studies investigated the difference in changes of VO_2_ after upper body exercise in interval and continuous endurance training programs [32,48]. 

### 3.3. Test Protocols

Table 1, Table 2 and Table 3 summarize all test protocols. The test protocols to determine VO_2peak_ and VO_2submax_ varied widely in the 27 studies evaluated. To determine VO_2peak ARM_, incremental (ITP) and discontinuous (DTP) test protocols were used. During ITPs on an arm crank ergometer (eight studies; [24,26,27,28,29,30,32,33]), the initial power output (W) ranged from 0 W [26] to 44.1 W [30], the increasing steps ranged from +5 W [32] to +30 W [27], the duration of the level was between 1 [24,28,29,32,33] and 3 min [30] and the revolutions per min differed from 50 to 70. Regarding ITPs on a wheelchair ergometer [42,43] or on a wheelchair on a motor-driven treadmill [39,41,44,45], the initial PO (W) was 0 W [42] or 8 W [43] or 20% of PO_peak_ [39,44,45], the increasing steps ranged from +10 W [42,43] or +10% of PO_peak_ [39,44,45], the duration of the level was between 1 [39,41,44,45] and 2 min [42,43] and the revolutions per min were set at 30 [42,43] or the velocity of the treadmill was constant at 1.39 m·s^−1^ [39,44,45]. On the handcycle, the initial PO (W) was 20 W [46,47] or 30 W [48], the increasing steps were +7 W [46,47] or +10 W [48], the duration of the level was 1 min and the revolutions per min were set at 70 [46,47,48] and/ or the velocity of the treadmill was constant at 1.39 m·s^−1^ [47] or 1.11 m·s^−1^ [46]. To determine submaximal parameters, such as VO_2submax_, longer-duration stages were used [16,17,18,24,26,27,30,33,34,35,38,39,43,44,45,46]. During submaximal arm crank ergometry, the duration of stages ranged from 5 [30] to 30 min [27] with an intensity that was always lower than 70% of VO_2peak_ [16,17,18,24,26,27,30,33,34,35]. To detect VO_2submax ARM_ during wheelchair ergometry, the investigators used a protocol with two 3 min levels at 20% and 40% PO_peak_ with a constant velocity (1.39 m·s^−1^) on the treadmill [38,39,44,45]. There was no extra protocol for submaximal parameters during handcycling. In 5 out of 27 studies (published before 1990), a discontinuous protocol (DTP) was used to detect maximal and submaximal parameters with work bouts (ranged from 4 to 5 min) and rest periods (ranged from 4 to 10 min) [31,35,36,37,40]. Not every protocol was explained sufficiently enough to provide all information reported in Table 1, Table 2 and Table 3.

### 3.4. Maximal UBET Responses

In Table 4 the pre- and post-training values for VO_2peak ARM_ and PO_peak ARM_ and the relative change (expressed in percentage from pre-training values) after training are listed. One of the 22 studies was excluded because of the same underlying data [38]. In 21 studies, VO_2peak ARM_ was measured; in total, 29 different training interventions were conducted with a VO_2peak ARM_ measurement. A total of 21 of the 29 different investigated training interventions reported a significant increase in VO_2peak ARM_ after training with an average of +16.4% (ranged from +7.2% [27] to 35.4% [24] with a mean standard deviation *(SD)* of 8.3%) [24,26,27,28,29,31,32,33,36,37,39,41,42,43,45,47,48]. Seven studies (reporting on eight training interventions) showed no significant increase in VO_2peak ARM_ after training (average: +1.83%; ranged from −1.9% [39] (T2) to +5.6% [32] (T1) with a *SD* of 3.1% [29,32,35,39,44,45,46]). A total of 16 studies collected PO_peak ARM_ data and, in total, 21 different training interventions were conducted to investigate changes in PO_peak ARM_ after training. A total of 19 of the 21 different training interventions showed a significant increase in PO_peak ARM_ after training with an average of +30.8% (ranged from +9.7% [32] to 63.6% [43] with a *SD* of 15.2% [26,27,28,32,33,35,39,42,43,44,45,46,47,48]). There was only one study with three different training modes which observed no significant increase in PO_peak ARM_ after upper body training (average: +5.2%; ranged from +2.1% to +8.8% with a *SD* of 2.8 [29]). Two studies which investigated maximal parameters presented their findings in a figure without providing exact data to analyze [36,40].

### 3.5. Submaximal UBET Responses

Since there is no consensus on the definition of submaximal performance within the included studies, a variety of different criteria and definitions for performance below the individual maximal workload had been applied. For the purpose of this systematic review, a pragmatic approach was taken and submaximal parameters were defined as workload at and below the ventilatory threshold (VT).

In Table 5, the pre- and post-training values for VO_2submax ARM_ and the relative change (expressed in percentages from pre-training values) after training are listed. One study [36] used a figure to visualize the training effects and did not report on exact data points. Therefore, 12 studies with 14 different training interventions out of the total 27 studies examined submaximal parameters as VO_2_ for analyses. A total of 4 out of 14 different training modes showed an increase in VO_2submax ARM_ after training [26,33,35,43]. Three out of these four studies showed a significant increase with an average of +30.3% (range: +17.8 [33] to +49.4% [43] with a *SD* of 11.9%) [26,33,43]. A total of 10 out of 14 different training interventions which focused on submaximal measures showed a decrease in VO_2submax ARM_ after training, with 7 studies showing a significant effect with an average of −17.2% (range: −12.9% [24,39] to −21.4% with a *SD* of 2.6%) [18,24,39,44,46]. Four out of four studies which also investigated power outputs at submaximal levels found a significant increase [26,33,35,43]. 

### 3.6. Transfer Effects

In Table 6 and Table 7, transfer effects, i.e., effects from trained (in this case: upper extremity, VO_2peak ARM_) to untrained (in this case: lower extremity, VO_2peak LEG_) musculature, are presented. A total of 12 out of 27 studies performed an incremental or discontinuous protocol on the cycle ergometer (11 studies [17,18,24,26,28,30,33,34,37,43,47]) or on the treadmill [31] before and after upper body training to analyze transfer effects in the untrained legs. A total of 9 out of these 12 studies investigated maximal parameters and of these 9 studies, 3 studies (Hill et al. [28], Loftin et al. [30] and Pogliaghi et al. [33]) showed a significant increase in VO_2peak LEG_ (average: +9.3%, ranged from +7% [30] to +13% [28] with a *SD* of 2.6%). The studies of Hill et al. [28] and Pogliaghi et al. [33] also showed a significant increase in PO_peak LEG_ (+10.2% [28] and +7.6% [33]). Focusing on submaximal transfer effects, as presented in Table 7, two studies (Lewis et al. [24] and McKenzie et al. [18]) reported significant decreases in VO_2submax LEG_ (−6.6% [24] and −8.5% [18]) and two studies (Pogliahi et al. [33] and Tordi et al. [43]) reported significant increases in VO_2submax LEG_ (+5.5% [33] and +21.4% [43]) with accompanying increases in PO_submax LEG_ (+9.1% [33] and +18.2% [43]) after upper body training. 

### 3.7. Training Devices for UBET

As shown in Table 1, Table 2 and Table 3, the training devices used in the studies differ markedly. There is no standard training device for arm cranking (A), wheelchair exercise (W) or handcycling (H). Arm cranking took place on a modified cycle ergometer [16,17,18,24,29,30,31,35,36,37], an arm crank ergometer [27,28,33] or an arm cycling device [26,32]. Wheelchair exercise was performed on a standardized wheelchair [38,39,42,44], a basketball wheelchair [41] on a motor driven treadmill, or on a wheelchair ergometer [40,43,45]. An attachable-unit handbike [46,47,48] was used for handcycling interventions. Further details on the training device are shown in Appendix A. 

## 4. Discussion

The purpose of this systematic review was to determine if UBET positively affects oxygen uptake as assessed by physical capacity in healthy individuals. The majority of the 27 final included studies reported an increase in VO_2peak ARM_ after UBET. A total of 21 of 29 interventions achieved a significant increase in VO_2peak ARM_ after training with an average increase of +16.4%. If the submaximal measures of oxygen uptake are considered (secondary outcomes), three studies observed a significant average increase of +30.3%, whereas seven studies showed a significant average decrease of −17.2% in VO_2submax ARM_. Three out of nine studies found significant transfer effects from trained arms to untrained legs (+9.3% in VO_2peak LEG_).

The current literature studying the effects of upper body exercises on VO_2_ in healthy people is still limited and only a few studies have been conducted with the specific goal to investigate outcomes on physical capacity after UBET in healthy individuals [18,28,29,31,32,33,35,36,37,39,43,44,45,46,47,48]. However, changes in VO_2_ after UBET were reported as exploratory endpoints in a number of studies [16,17,24,26,27,30,34,38,40,41,42]. One of the main challenges of most studies is the low number of participants included in the intervention and control groups, if present (the number of participants per study ranged between 5 and 20 with a mean value of only 9 subjects per study). This fact leads to low statistical power with a low detection rate for significant results on the one hand and potentially inflated effect sizes on the other hand. In addition, heterogeneity of the samples lowers statistical power even further by increasing variance of no interest and lowering the signal-to-noise ratio. This heterogeneity of the samples is caused by variations in age, gender and training status, which are expected to impact the training effects. Five studies only recruited females [30,35,40,46,47] and two studies reported on mixed populations [28,32] as sample. Therefore, it is difficult to interpret and compare the current results of UBET and apply them equally to men and women. In addition to heterogeneity introduced by inter-individual variability, different training modes (length, intensity, duration, test device) and different maximal and submaximal exercise tests to measure VO_2_ (i.e., ITP or DTP, initial power, power increments after each step, duration of the steps) also affect the strength of the conclusions drawn by this current review. Additionally, potential selection bias is likely because older studies (dating back until 1990) did not report on the recruitment strategies (e.g., general population vs. athletic individuals). 

### 4.1. Maximal UBET Responses

In the next section, possible factors that may influence maximum measures according to UBET are presented and discussed. 

#### 4.1.1. Effects of Training Intensity

Almost all studies clearly showed that UBET has a positive effect on the physical capacity, as can be seen from the improvements in VO_2peak ARM_ (+16.4%) and PO_peak ARM_ (+30.8%). These overall findings are comparable with the results of Hettinga et al. [47]. This study showed that UBET based on ACSM guidelines (7 weeks of handcycling training, 3 × 30 min/week at 65% heart rate reserve (HRR)) led to local adaptations improving handcycling performance in healthy young females (VO_2peak ARM_ increased by +18.1% and PO_peak ARM_ increased by +31.4%). Importantly, the ACSM guidelines refer to lower body training exercise with larger muscle mass such as walking, running or cycling [49]. Hence, these guidelines can be used as a basis to design UBET for females with limited active muscle mass [47]. However, when the intensity of the training is low and the duration of the training session is less than one hour, quantitative improvements in VO_2peak ARM_ have been observed, but these changes were not significant (Table 4, [39,44,45,46]). Abonie et al. [46] applied the same training (length, duration, test device, test protocol) as Hettinga et al. [47], but with lower intensity (30% HRR versus 65% HRR). This difference led to no significant improvement in VO_2peak ARM_ in healthy young females (+7% versus +18.1%). An increase of +15.8% in VO_2peak ARM_ was also measured after 8 weeks of continuous UBET at 72% VO_2peak ARM_ in male subjects [26]. After 10 weeks of UBET with an intensity of 85% HR_peak_ during 6 × 4 min work bouts [31] or with an intensity of 180 bpm over 10 min [37], similarly high improvements in VO_2peak_ were achieved (+16.4% and +17%). Thus, it can be concluded that different UBET lead to comparable changes in VO_2peak ARM_. 

#### 4.1.2. Effects of Baseline Fitness and Length of Training Program

Hill et al. [28] and Pogliaghi et al. [33] studied older persons (≥60 years of age) and showed the highest increases in VO_2peak_ after UBET in healthy persons (+24.1% and +22.8%). Maximum oxygen uptake (VO_2max_) decreases with age [50] and, therefore, the subjects had a relatively low fitness level and low initial value in their VO_2peak_. This might explain the large improvements in VO_2peak ARM_ in older male subjects compared to trained handcyclists. The long training period (over 12 weeks) in the study of Pogliaghi et al. [33] may also be a reason for the strong increase in VO_2peak ARM_ after UBET. The greatest increase in VO_2peak ARM_ of +35.4% (from 1.64 L*min^−1^ to 2.22 L*min^−1^) was found in the study of Lewis et al. [24] after 11 weeks of UBET on an arm crank ergometer with an intensity of 75–80% VO_2peak ARM_. Once again, the initial low fitness level and the length of the training period may have been the reason for the strong improvement in VO_2peak ARM_.

#### 4.1.3. Effects of Motor Learning of UBET

For all observed improvements in VO_2peak ARM_, it is important to note that none of these 27 studies performed familiarization tests on the arm crank device, wheelchair or handcycle. Therefore, a learning effect between pre-test and post-test must also be assumed. For healthy persons, training on the arm crank ergometer, wheelchair or handcycle might have been unfamiliar since it is not a daily workout routine such as walking or cycling. The strong increases in PO_peak ARM_ up to +42.1%–+63.6% [38,42,43,45,48] can be at least partly explained by a learning effect from pre- to post-test in addition to an improvement in physical capacity due to local and peripheral adaptations. The greatest improvement in PO_peak ARM_ of +63.6% (from 66 W in pre- to 108 W in post-test) was found in the study of Tordi et al. [43], accompanied by an increase of +29.3% in VO_2peak ARM_. The participants in this study conducted Square-Wave Endurance Test (SWEET) training over 5 weeks. One SWEET session consisted of nine consecutive periods of 5 min including 4 min “base” work followed by 1 min “peak” work (close to HR_max_). When the initial fitness level of subjects is considered, with these achieved peak VO_2_ and PO values after a training intervention, it can be said that SWEET is an effective UBET protocol to improve upper body capacity in a short time [42,43]. However, future studies should also perform familiarization tests to exclude learning effects from pre- to post-test. 

### 4.2. Submaximal UBET Responses

Focusing on the outcome of the submaximal values of VO_2_, four interventions reported an increase in VO_2_ and, in ten, a reduction. These contradicting results may be explained by the variable use of different test protocols used in these studies. Whereas studies which determined an increase in VO_2_ measured at the VT, the studies with a reduction in VO_2_ at predefined workloads measured below the VT. However, both outcomes can be interpreted as having a positive training effect. 

The significant increases in VO_2submax ARM_ were determined during higher workloads, and specifically for workloads located at the VT (Table 1, Table 2 and Table 3, test protocol for VO_2submax ARM_) [26,33,43]. The increase in VO_2submax ARM_ at VT, therefore, can be interpreted as a positive training effect after UBET since it indicates a higher availability of oxygen to the subjects at intense levels of exertion. 

Significant decreases in VO_2submax ARM_, on the other hand, were observed in exercise tests at lower workloads (Table 1, Table 2 and Table 3, test protocol for VO_2submax ARM_: at 70% of VO_2peak ARM_ [24]; at 63 W and 83 W [18]; at 20% and 40% PO_peak ARM_ [39,44]; at 27 W, 41 W and 65 W [46]). The decreases in VO_2 submax ARM_ imply that less oxygen is required to perform at the same intensity compared to the baseline. The body or, more precisely, the cardiovascular system of participants, likely works more economically at the same intensity than before UBET. The largest drop in oxygen uptake during submaximal workloads was identified in the study of de Groot et al. [39]. In this study, three different training designs with different intensities (70 min at 30% HRR, 30 min at 30% HRR and 30 min at 70% HRR) were compared. The training design with the longest duration and lowest intensity achieved the strongest decrease of −21.4% in VO_2submax ARM_, but there were no significant differences in changes in submaximal physical capacity parameters found between the three training groups. Thus, training with an intensity lower than recommended by the guidelines of ACSM [49] (proposing a minimal training intensity of 40% HRR) led to an improvement in aerobic capacity. Submaximal parameters also play a decisive role in the goal of achieving the highest possible fat oxidation rate (Fat_max_). In submaximal ranges, fat metabolism reaches maximum values and then drops again, whereas carbohydrate metabolism continues to increase at or beyond VT [51,52]. According to recent studies, Fat_max_ in UBET is lower compared to that in cycling (0.44 +/− 0.24 g*min^−1^ versus 0.77 +/−0.31 g*min^−1^) and has occurred at lower intensities (53 +/− 21% versus 67 +/−18% VO_2peak ARM_) [53]. Given that the highest fat oxidation was found during UBET with very low loads, an improvement in fat metabolism is associated with an improvement in performance capacity and overall health [54,55,56,57]. If evidence in future studies confirms this finding, UBET at submaximal intensities could be an alternative therapeutic approach for obese patients with lower extremity joint pain, prohibiting classical training such as walking or cycling.

The influence of the test protocol (variations in workload (W) and durations of the steps) on the outcome can be observed by the variability of submaximal VO_2_ parameters (Table 5, increases and decreases in VO_2submax ARM_). However, due to the large variation of the test protocols, interpretation of the improvements in submaximal (and maximal) exercising tests proved to be difficult. For future studies, the use of standardized protocols for men and women should be implemented to measure physical changes after UBET to enable the comparison of study results and inform guidelines for evidence-based UBET.

### 4.3. Transfer Effects

Most studies that have analyzed the effects of UBET describe positive changes in the trained muscles. Therefore, local adaptations in the trained musculature are likely to have taken place. The physiological basis of these improvements (an increase in oxygen utilization in the trained muscles) needs to be further analyzed in future research, e.g., by including muscle biopsies. Factors that might influence the increase in oxygen utilization are capillary density, increased activity of oxidative enzymes or decreases in the activity of glycolytic enzymes, an increase in mitochondrial density, a conversion of type IIb muscles fibers to type IIa or a combination of these aspects. Thus, the training effects of UBET are mainly localized and due to changes in propulsion technique, which is consistent with the results of various studies that could not find transfer effects in untrained legs after UBET (Table 6 and Table 7, [16,26,31,37,43,47]). Conversely, three studies showed significant improvements in VO_2peak LEG_ after UBET without training the leg muscles [28,30,33]. The fact that significant transfer effects can also take place in submaximal ranges is shown by four studies [18,24,33,43]. Transferable effects to untrained muscles are in agreement with the findings of Saltin et al. [15] and Clausen et al. [16] who, in leg training (one-legged training or two-legged training), saw an enhancement in the central circulation which could be transferred to non-trained muscles. These are promising findings and particularly interesting for athletes who cannot continue training due to lower extremity injuries. Evidence-based training that improves leg performance without moving the legs would likely result in faster recovery and fewer major training reductions in healthy persons. Thus, there is a great need for future research to examine this highly interesting approach in sports therapy and sports science. 

#### 4.3.1. Effects of Baseline Fitness Level and Age 

Changes in central adaptations (an increase in cardiac output and oxygen delivery to the untrained muscles) after UBET were found in elderly participants (>60 years) with a low initial fitness in the studies of Pogliaghi et al. [33] and Hill et al. [28]. In the study of Pogliaghi et al. [33], male participants completed an intense and continuous training (30 min at VT, 3d*wk-1) program over 12 weeks. In this case, the low initial fitness levels and long training periods could be decisive for achieving significant transfer effects of +9.1% in VO_2peak LEG_ and +7.6% in PO_peak LEG_. Hill et al. [28] confirmed these results with a mixed-gender study cohort and a shorter training period (5 versus 12 weeks of training) and found an increase of +13.8% in VO_2peak LEG_ and +10.2% in PO_peak LEG_. Loftin et al. [30] explained these cross-transfer effects from arms to legs as indirect evidence of central adaptations, which become apparent in an improved cardiac output and stroke volume. 

#### 4.3.2. Gender

The studies by Hettinga et al. [47] and Loftin et al. [30] are the only training studies that have investigated transfer effects from arms to legs in women. There was an significant increase in VO_2peak LEG_ after arm training in the study by Loftin et al. [30]. A mixed sample was examined in the study by Hill et al. [28]. They found a significant improvement in VO_2peak LEG_ after arm training but, unfortunately, the sample was not analyzed by gender. Thus, based on the available material of studies, it is not possible to draw conclusions which elucidate how sex influences the magnitude or presence of transfer effects.

#### 4.3.3. Effects of Training Design

The training design can differ in terms of training method (continuous or interval), length (days/ week), intensity and duration (min). Considering the method of training, whether it was a continuous training or an interval training, it is striking that two of three studies (Hill et al. [28] and Pogliaghi et al. [33]) showing an significant increase in VO_2peak LEG_ were continuous arm training protocols. Within the results of submaximal transfer effects, two studies (McKenzie et al. [18] and Tordi et al. [43]) conducted interval training protocols and two studies conducted continuous training protocols (Lewis et al. [24] and Pogliaghi [33]). Therefore, the training method does not seem to be a decisive factor for the presence of transfer effects. It should be emphasized that the training studies that found significant transfer effects were longer than 5 weeks and took place at moderate to high intensities. However, it must be mentioned that, to date, there has been no study that has investigated transfer effects following several weeks of low-intensity UBET. 

#### 4.3.4. Effects of Training Device

Most studies (10 out of 12) that have looked at transfer effects from arms to legs have performed their training on an arm crank ergometer. Three of them showed a significant increase in VO_2peak LEG_ after exercising the arms and three more described significant changes in VO_2submax LEG_. Only the study by Tordi et al. [43] showed a significant increase in VO_2submax LEG_ after training on a wheelchair ergometer. There is also only one study by Hettinga et al. [47] with an underlying training intervention on a handcycle device. However, no transfer effects from arms to legs were found in this study. The fact that most studies have used an arm ergometer limits any conclusion as to whether the device impacts the presence or magnitude of transfer effects. 

To summarize, these data of significant transfer effects (Table 6 and Table 7, [24,28,30,33]) suggest that the increase in peak exercise is training mode-specific, partially locally in the trained muscles as well as systemically (e.g., to the untrained leg muscles). It must be clearly stated that transfer effects from arms to leg have so far only occurred in populations with a low initial fitness. Since individuals’ responses to training and the capacity to adapt have been reported to vary [58], subject information, such as the initial fitness level or gender, should always be taken into account. 

## 5. Conclusions

This systematic review provides a comprehensive overview of the effects of UBET on aerobic fitness changes in healthy people. Table 8 shows a summary of results where the analyzed studies were divided into three groups based on the intensity of the training sessions, i.e., the outcome of UBET (low: 30–50% HRR, HR_peak_, VO_2peak_; moderate: 60–80% HRR, HR_peak_, VO_2peak_; and high: >80% HRR, HR_peak_, VO_2peak_. Grouping is made for easier comparability of studies and may differ from ACSM guidelines). Different forms of UBET (various training designs with different training devices) led to improvements in the trained arm muscles (VO_2peak/submax ARM_ and PO_peak ARM_). For transfer effects from arms to legs, the evidence was less conclusive and merits further investigation to clarify the physiological basis of the effect. Given the research examining UBET in healthy individuals, conclusions on how to apply UBET in healthy persons are also reasonable based on this systematic review. Because UBET studies used a variety of modes (i.e., an arm crank ergometer, a wheelchair ergometer or a handcycle), the recommendations below are applicable to different modes of UBET.

Recommendations for UBET based on the results of this systematic review are: UBET studies are, in general, of small sample sizes and may, therefore, fail to detect potential training effects and may at the same time be at risk of overestimating training effects;UBET leads to the largest effects in improving physical capacity when training is performed for longer than 5 weeks at an intensity >70% of VO_2peak ARM_ or HRR _ARM_;The SWEET training design was found to be very effective;ACSM guidelines for larger muscle masses (legs) (7 weeks of training, 3 × 30 min/ week at 65% HRR) can also be used on small muscle groups (arms) to improve aerobic fitness;Low-intensity training (30% HRR) improves (sub)maximal VO_2_ parameters and plays a decisive role in fat metabolism training;UBET is a complementary and useful workout but does not replace whole-body exercises;Since the majority of subjects were males (more than 85%), more studies need to be conducted in females to understand if the same effects occur in females and whether UBET recommendations are sex-specific.

## Figures and Tables

**Table 1 biology-12-00355-t001:** Characteristics of studies evaluating the effects of arm crank exercise (ACE) on aerobic performance.

Study (Author, Year)	Sample			Control Group/ Leg Group/SCII			Training Status	Training Design				Physical Capacity Outcomes	Other Outcomes	Test Protocol	
	n	Sex	Age (y)	n	Sex	Age (y)		Length	Intensity	Duration (min)	Test Device			VO_2peak ARM_	VO_2submax ARM_
**Bhambhani** **1991 [26]**	8	♂	38	8 LG	♂	41	Physically active, no specific arm training	3 d × wk^−1^ 8 wk	Con, 72% VO_2peak_	30	A Arm cycling device	VO_2peak_ PO_peak_	Peak and submax VO_2_, PO, HR, V_E,_ RER, O_2pulse,_ RPE for arm and leg exercise	ITP: Initial PO: 0W; +12.5W every 2 min; 50 rpm	6min at VT; 50 rpm
**Clausen 1973 [16]**	5	♂	23	5 LG	♂	24	Physically active, no specific training	5 d × wk^−1^ 5 wk	I- 4 × 5 min at 170 bpm	4 × 5	A Modified cycle ergometer	not measured	Submax VO_2_, LA, RER, HR		2 × 15 min at 130 and 170 bpm
**El-Sayed 2004 [27]**	7	n.d.	32	5 SCII	n.d.	31	Untrained, sedentary lifestyle	3 d × wk^−1^ 12 wk	Con, 65% VO_2peak_	30	A Arm crank ergometer	VO_2peak_ PO_peak_	Peak HR, VE; submax platelet aggregation	ITP: Initial PO: 30W; +30 W every 2 min; 60–65 rpm	30 min at 60–65% VO_2peak_; 60–65 rpm
**Hill 2018 [28]**	10	♀/ ♂	66	10 LG	♀/ ♂	66	Physically active, no specific training	3 d × wk^−1^ 6 wk	Con, 50–70% PO_peak_	20–45	A Arm crank ergometer	VO_2peak_ PO_peak_	Peak HR, V_E_, RER, RPE for arm and leg exercise	ITP: Initial PO: 25W; +10 every min; 60 rpm	
**Klausen 1974 [17]**	5	♂	23	5 LG	♂	24	Physically active, no specific training	5 d × wk^−1^ 5 wk	I, 4 × 5 min at 170 bpm	4 × 5	A Modified cycle ergometer	not measured	Submax LA, VO_2_, RER, HR		2 × 15 min at 130 and 170 bpm
**La Monica 2019 [29]**	T1: 11	♂	23	8 CG	♂	24	Physically active, no specific arm training	3 d × wk^−1^ 2 wk	Int, All-out sprints using 0.05kg × kg^−1^ body mass loading	4 × 10 s (2 min rest)	A Modified cycle ergometer	VO_2peak_ PO_peak_	Wingate Test (CP, W‘, PP, MP, TW), EMG_FT_	ITP: Initial PO: 30W; +10W every min; 50 rpm	
	T2: 11	♂	22							4 × 10 s (4 min rest)					
	T3: 10	♂	23							4 × 30 s (4 min rest)					
**Lewis 1980 [24]**	5	♂	20	5 LG	♂	22	Physically active, no specific training	4 d × wk^−1^ 11 wk	Con, 75–80% VO_2peak_	30	A Modified cycle ergometer	VO_2peak_	Peak and submax VO_2_, HR, V_E_, V_E/_VO_2_, RPE for arm and leg exercise	ITP: Initial PO: 25W; +17W every min, 70 rpm	2 × 10min at 70% VO_2peak_; 60 rpm
**Loftin 1988 [30]**	19	♀	22	19 CG	♀	24	n.d.	4 d × wk^−1^ 5 wk	Int, 2 × 70% HRR, 2 × 80% HRR, 2 × 90% HRR	6–4	A Modified cycle ergometer	VO_2peak_	Peak V_E_, Q, HR, SV, (a-v) O_2diff_, for arm and leg exercise	ITP: Initial PO: 44.1W; +14.7 every 3 min, 60 rpm	5 min at 14.7 W and 5 min at 29.4 W
**Magel 1978 [31]**	9	♂	24	7 CG	♂	23	n.d.	3 d × wk^−1^ 10 wk	Int, 85% HR_peak_	6 × 4	A Modified cycle ergometer	VO_2peak_	Peak and submax VO_2_, VE_,_ RER, Q, HR, SV, (a-v) O_2diff._ for arm and leg exercise (running)	DTP: 4 min work and 10 min rest; Initial PO: 0W, 40 rpm; resistance was increased to 1.0 kg (240 kg × m^−1^ × min^−1^) and then by 0.5 kg (120 kg·m^−1^·min^−1^)	
**McKenzie 1978 [18]**	7	♂	20	8 LG	♂	20	Untrained, sedentary lifestyle	5 d × wk^−1^ 5 wk	Int, close to 180 bpm	40–45 (work bouts between 30 s and 2 min)	A Modified cycle ergometer	not measured	Submax VO_2,_ HR, LA, total exercise time		8–10 min at 63 and 83 W, 50 rpm
**Pinto 2019 [32]**	T1: 10	♀/ ♂	24				physically active, no specific training	2 d × wk^−1^ 6 wk	Con, 80% HR_peak_	20	A Arm cycling device	VO_2peak_ PO_peak_	Peak HR, RPE; LA	ITP: Initial PO: ♀: 20 W; +10 W/min up to 50 W, then +5 W/min ♂: 20 W; +20 W/min up to 60 W, then +10 W/min; >50 rpm	
	T2: 10	♀/ ♂	23						Int, 90% HR_peak_	20 (1 min on, 1 min off)					
**Pogliaghi 2006 [33]**	6	♂	68	6 LG	♂	66	Untrained, sedentary lifestyle	3 d × wk^−1^ 12 wk	C at VT	30	A Arm crank ergometer	VO_2peak_ PO_peak_	Peak and at VT (submax) VO_2_, PO, V_E,_ RER, HR, O_2pulse_	ITP: Initial PO: 40 W; +5 W/min	No extra submax test protocol; VT was detected by visual inspection based on ventilatory equivalents and end-tidal fractions of O_2_ and CO_2_
				6 CG	♂	73									
**Rasmussen 1975 [34]**	5	♂	23	5 LG	♂	24	not described	5 d × wk^−1^ 5 wk	Int, Intermittent, maximal and dynamic exercise	60	A Not described	not measured	Submax VO_2_, V_E,_ V_T,_ F_resp_, HR, Sa_O2_, Sv_O2_		15 min at a moderate and at a heavy submaximal workload
**Sedlock 1988 [35]**	6	♀	25	4 CG	♀	23	Untrained, sedentary lifestyle	3 d × wk^−1^ 5 wk	Int. 85% HR_peak_	4 × 4	A Modified cycle ergometer	VO_2peak_ PO_peak_	Peak HR, LA and submax VO_2,_ HR, LA, Q, SV	DTP: Initial PO: 12 W; PO levels were progressively increased by 12 W with 4 min rest periods, 50 rpm	10 min at 70% VO_2peak_
**Simmons 1978 [36]**	10	♂	25				No regular physical exercise	2 d × wk^−1^ 4 wk	Con, 80% VO_2peak_	30	A Modified cycle ergometer	VO_2peak_ PO_peak_	Peak and submax GE, Q, HR (*graphic)	DTP: 5 min work bouts, 5 min rest; 4 workloads were selected so that each individual reached his maximum oxygen intake; 60 rpm	
**Stamford 1978 [37]**	8	♂	20	9 LG	♂	19	Untrained, sedentary lifestyle	3 d × wk^−1^ 10 wk	Con, 180bpm	10	A Modified cycle ergometer	VO_2peak_	Peak HR, RER, V_E_	DTP: 5 min work bouts, 10 min rest; initial resistance: 1.5 kg, +0.5 kg/ work bout; 60 rpm	

ACE, arm crank exercise; T1–3, training group 1–3; LG, leg group; CG, control group; SCII, spinal cord-injured individuals; n.d., not described; Int, interval; Con, continuous; HRR, heart rate reserve; VO_2_, oxygen uptake; bpm, beats per min; PO, power output; VT, ventilatory threshold; A, arm crank ergometer; submax, submaximal; HR, heart rate; V_E_, ventilatory efficiency; RER, respiratory exchange ratio; O_2_, oxygen; RPE, rate of perceived exertion; LA, lactate; CP, critical power; W’, anaerobic power; MP, mean power; TW, total work; EMG_FT_, electromyographic fatigue threshold; Q, cardiac output; SV, stroke volume; (a–v) O_2diff_, arteriovenous oxygen difference; V_T_, tidal volume; F_resp_, respiratory frequency; Sa_o2_, arterial oxygen saturation; Sv_o2_, oxygen saturation and oxygen tension in venous blood; GE, gross efficiency; ITP, incremental test protocol; DTP, discontinuous test protocol; rpm, revolutions per min.

**Table 2 biology-12-00355-t002:** Characteristics of studies evaluating the effects of wheelchair exercise (WCE) on aerobic performance.

Study (Author, Year)	Sample			Control Group/ Leg Group/ SCII			Training Status	Training Design				Physical Capacity Outcomes	Other Outcomes	Test Protocol	
	n	sex	age (y)	n	sex	age (y)		Length	Intensity	Duration (min)	Test Device			VO_2peak ARM_	VO_2submax ARM_
**De Groot 2008 [38]**	14	♂	24	7		23	Physically active, no specific arm training	3 d × wk^−1^ 7 wk	Con, 30% HRR	70	W Standardized wheelchair		Submax VO_2,_ HR; GE		2 × 3 min at 20% and 40% PO_peak_; 1.39 m·s^−1^
**De Groot 2013 [39]**	T1: 14	♂	24				Physically active, no specific arm training	3 d × wk^−1^ 7 wk	Con, 30% HRR	70	W Standardized wheelchair	VO_2peak_ PO_peak_	Isometric strength, sprint power, peak HR; submax VO_2,_ HR; GE	ITP: Initial PO: 20% PO_peak_; +10% estimated PO_peak_ every min; 1.39 m·s^−1^	2 × 3 min at 20% and 40% PO_peak_; 1.39 m·s^−1^
	T2: 10	♂	23						Con, 30% HRR	30					
	T3: 13	♂	22						Con, 70% HRR	30					
**Glaser 1981 [40]**	7	♀	21	6 CG	♀	22	n.d.	3 d × wk^−1^ 7 wk	Int, 80% HR_peak_	3 × 4	W Wheelchair ergometer	VO_2peak_ PO_peak_ (*graphic)	Peak and submax VO_2_, V_E_, HR	DTP: Exercising at each PO level (30, 60, 90, 120, 150 kpm·min^−1^) for 4 min; 30 rpm	
**Goosey-Tolfrey 2011 [41]**	T1: 8	n.d.	20	6 CG	n.d.	20	Physically active, no specific arm training	3 d × wk^−1^ 3 wk	Listened to 170bpm	1 × 4	W Basketball wheelchair	VO_2peak_	Peak HR, RPE; GE; propulsion technique	ITP: 0.1–0.2 m·s^−1^ increments every min	
	T2: 8	n.d.	20						Listened to 125bpm						
**Grange 2002 [42]**	7	♂	27	7 SCII	♂	35	Physically active, no specific training	3 d × wk^−1^ 6 wk	Int, 4min at VT and 1 min at PO_peak_	9 × 5 (SWEET)	W Standardized wheelchair	VO_2peak_ PO_peak_	Peak RPE, HR	ITP: Initial PO: 0 W; +10 W every 2 min; 30 rpm	
**Tordi 2001 [43]**	5	♂	23	5 LG	♂	23	Physically active, no specific training	3 d × wk^−1^ 6 wk	Int, 4min at VT and 1 min at PO_peak_	9 × 5 (SWEET)	W Wheelchair ergometer	VO_2peak_ PO_peak_	Maximal and at VT VO_2_, V_E,_ HR, O_2pulse_	ITP: Initial PO: 8 W; +10 W every 2 min; 30 rpm	VT was detected by visual inspection based on ventilatory equivalents and end-tidal fractions of O_2_ and CO_2_
				5 CG	♂	23									
**Van den Berg 2010 [44]**	10	♂	23	15 CG	♂	23	Not actively engaged in sports over the last year	3 d × wk^−1^ 7 wk	Con, 30% HRR	30	W Standardized wheelchair	VO_2peak_ PO_peak_	F_iso_, peak HR, submax VO_2_, HR, RER; GE	ITP: Initial PO: 20% PO_peak_; +10% PO_peak/_ min; 1.39m·s^−1^	2 × 3 min (20% PO_peak_ and 40% PO_peak_) at 1.39m·s^−1^
**Van der Woude 1999 [45]**	T1: 9	♂	23	8 CG	♂	22	Not actively engaged in sports over the last year	3 d × wk^−1^ 7 wk	Con, 50% HRR	30	W Wheelchair ergometer	VO_2peak_ PO_peak_	F_iso_, sprint power, peak HR, RER, submax HR; GE	ITP: Initial PO: 20% PO_peak_; +10% PO_peak/_ min; 1.39 m·s^−1^	2 × 3 min (20% PO_peak_ and 40% PO_peak_) at 1.39 m·s^−1^
	T2: 10	♂	23						Con, 70% HRR						

WCE, wheelchair exercise; T1–3, training group 1–3; LG, leg group; CG, control group; SCII, spinal cord-injured individuals; n.d., not described; Int, interval; Con, continuous; SWEET, Square-Wave Endurance Exercise Test; HRR, heart rate reserve; VO_2_, oxygen uptake; bpm, beats per min; PO, power output; VT, ventilatory threshold; W, wheelchair; submax, submaximal; HR, heart rate; V_E_, ventilatory efficiency; RER, respiratory exchange ratio; O_2_, oxygen; RPE, rate of perceived exertion; LA, lactate; Q, cardiac output; SV, stroke volume; GE, gross efficiency; F_iso_, maximal isometric strength; ITP, incremental test protocol; DTP, discontinuous test protocol; rpm, revolutions per min.

**Table 3 biology-12-00355-t003:** Characteristics of studies evaluating the effects of handcycle exercise (HCE) on aerobic performance.

Study (Author, Year)	Sample			Control Group/ Leg Group/ SCII			Training Status	Training Design				Physical Capacity Outcomes	Other Outcomes	Test Protocol	
	n	Sex	Age (y)	n	Sex	Age (y)		Length	Intensity	Duration (min)	Test Device			VO_2peak ARM_	VO_2submax ARM_
**Abonie 2021 [46]**	9	♀	21	10 CG	♀	21	Physically active, no specific training	3 d × wk^−1^ 7 wk	Con, 30% HRR	30	H Attachable- unit handbike	VO_2peak_ PO_peak_	Peak HR, RER, V_E_ RPE; submax VO_2_, HR, RER, V_E,_ RPE; GE	ITP: Initial PO: 20 W; +7 W every min, 1.11 m·s^−1^	No extra submax protocol: 2nd, 4th and 6th stages of incremental test (PO of 27 W, 41 W and 65 W) were evaluated for submaximal performance
**Hettinga 2016 [47]**	11	♀	22	11 CG	♀	21	Physically active, no specific training	3 d × wk^−1^ 7 wk	Con, 65% HRR	30	H Attachable- unit handbike	VO_2peak_ PO_peak_	Peak HR, V_E,_ RPE, RER for arm and leg exercise	ITP: Initial PO: 20 W; +7 W every min; 1.39 m·s^−1^, 70 rpm	
**Schoenmakers 2016 [48]**	T1: 8	♂	21	8 CG	♂	23	Physically active, no specific training	3 d × wk^−1^ 7 wk	Con, 66% HRR	30	H Attachable- unit handbike	VO_2peak_ PO_peak_	Peak V_E,_ RER, HR	ITP: Initial PO: 30 W; +10 W every min; 70 rpm	
	T2: 8		23						Int, 85% HRR	4x4					

HCE, handcycle exercise; T1–2, training group 1–2; CG, control group; SCII; Int, interval; Con, continuous; HRR, heart rate reserve; VO_2_, oxygen uptake; H, handcycle; PO, power output; submax, submaximal; HR, heart rate; V_E_, ventilatory efficiency; RER, respiratory exchange ratio; O_2_, oxygen; RPE, rate of perceived exertion; GE, gross efficiency; ITP, incremental test protocol; DTP, discontinuous test protocol; rpm, revolutions per min.

**Table 4 biology-12-00355-t004:** Mean change in VO_2peak ARM_ and PO_peak ARM_ between pre- and post-test.

Study (Author, Year)	Test Device	n	VO_2peak ARM_						PO_peak ARM_		
			Pre-Test		Post-Test		% Change		Pre-Test (W)	Post-Test (W)	% Change
			(L × min^−1^)	(ml × min^−1^ × kg^−1^)	(L × min^−1^)	(ml × min^−1^ × kg^−1^)	(L × min^−1^)	(ml × min^−1^ × kg^−1^)			
**Bhambhani** **1991 [26]**	A	8	2.72	32.0	3.15	37.5	+15.8 *	+17.2 *	96.4	108.7	+12.8 *
**El Sayed 2004 [27]**	A	7	1.81	24.1	1.94	26.2	+7.2 *	+8.7 *	168	185	+10.1 *
**Hill 2018 [28]**	A	10	1.12	17	1.39	22	+24.1 *	+29.4 *	51	65	+27.5 *
**La Monica 2019 [29]**	A	T1: 11	2.53	29.0	2.53	29.6	0	+2.1	140	143	+2.1
		T2: 11	2.60	27.9	2.58	28.9	−0.8	+3.6	130	136	+4.6
		T3: 10	2.21	28.6	2.36	31.1	+6.8 *	+8.7	125	136	+8.8
**Loftin 1988 [30]**	A	19					+33 *	+32 *			
**Lewis 1980 [24]**	A	5	1.64		2.22		+35.4 **				
**Magel 1978 [31]**	A	9	2.69	33.9	3.13	39.3	+16.4 **	+15.9 **			
**Pinto 2019 [32]**	A	T1: 10	1.8	27.2	1.9	28.3	+5.6	+4.0	72	79	+9.7 *
		T2: 10	2.2	33.5	2.5	38.3	+13.6 **	+14.3 **	89	101	+13.5 *
**Pogliaghi 2006 [33]**	A	6	1.62	22.0	1.99	26.8	+22.8 *	+21.8 *	87	106	+21.8 *
**Sedlock 1988 [35]**	A	6	1.38		1.43		+3.6		60.4	68.8	+13.9 *
**Simmons 1971 [36]**	A	10	2.76		2.99		+8.3 *		figure		
**Stamford 1978 [37]**	A	8	2.82	36.9	3.3	44	+17 **	+19.2 **			
**De Groot 2008 [38]**	W	14	1.75		1.95		+11.4 *		43.7	66.7	+52.6 *
**De Groot 2013 [39]**	W	T1: 14	1.75		1.95		+11.4 *		43.7	66.7	+52.6 *
		T2: 10	2.13		2.09		−1.9		56.5	75.6	+33.8 *
		T3: 13	1.80		2.0		+11.1 *		52.9	79.0	+49.3 *
**Goosey-Tolfrey 2011 [41]**	W	8	1.73		1.99		+15 *				
		8	1.73		1.89		+9.2 *				
**Grange 2002 [42]**	W	7		34.7		37.5	+8.3 *		61.6	89.3	+45 *
**Tordi 2001 [43]**	W	5		30.4		39.3		+29.3 **	66	108	+63.6 ***
**van den Berg 2010 [44]**	W	9	2.13	27.8	2.09	27.2	−1.9	−2.2	56.4	75.6	+34 *
**van der Woude 1999 [45]**	W	T1: 9	1.79		1.88		+5		56.3	73	+29.7 *
		T2: 10	1.85		2.03		+9.7 *		57.9	82.3	+42.1 *
**Abonie 2021 [46]**	H	9	1.60	26.4	1.68	27.5	+5	+4.2	81.1	97.4	+20.1 *
**Hettinga 2016 [47]**	H	11	1897	28.3	2240	33.2	+18.1 *	+17.3 *	89	117.4	+31.9 *
**Schoenmakers 2016 [48]**	H	T1: 8		33.2		36.5	+9.9 *		128.9	169	+31.1 *
		T2: 8		34.3		41.9	+22.2 *		133.2	191.3	+43.6 *

A, arm crank ergometer; W, wheelchair; H, handcycle; T1–3, training group 1–3; VO_2_, oxygen uptake; PO, power output; *, **, *** significance of the difference between pre-training and post-training at the level of *p* < 0.05, 0.01, 0.001, respectively.

**Table 5 biology-12-00355-t005:** Mean change in VO_2submax ARM_ and PO_submax ARM_ between pre- and post-test.

Study (Author, Year)	Test Device	n	VO_2subamx ARM_						PO_submax ARM_		
			Pre-Test		Post-Test		% Change		Pre-Test (W)	Post-Test (W)	% Change
			(L × min^−1^)	(ml × min^−1^ × kg^−1^)	(L × min^−1^)	(ml × min^−1^ × kg^−1^)	(L × min^−1^)	(ml × min^−1^ × kg^−1^)			
**Bhambhani** **1991 [26]**	**A**	8	1.18	13.8	1.46	17.4	+23.7 *	+26.1 *	50.7	59.8	+17.9 *
**Clausen 1973 [16]**	**A**	5	1.25		1.15		−8				
**Klausen 1974 [17]**	**A**	5	1.25		1.15		−8				
**Lewis 1980 [24]**	**A**	5	1.24		1.08		−12.9 **				
**Magel 1978 [31]**	**A**	9	1.67		1.64		−1.8				
**McKenzie 1978 [18]**	**A**	7	1.36		1.12		−17.6 **				
**Pogliaghi 2006 [33]**	**A**	6	1.07		1.26		+17.8 *		60	70	+16.7 *
**Rasmussen 1975 [34]**	**A**	5	1.76		1.64		−6.8				
**Sedlock 1988 [35]**	**A**	6	0.82		0.85		+3.7		39.5	44.5	+12.7 *
**Simmons 1975 [36]**	**A**	10	graphic								
**De Groot 2008 [38]**	**W**	14	1.17		0.92		−21.4 *				
**De Groot 2013 [39]**	**W**	T1: 14	1.17		0.92		−21.4 *				
		T2: 10	1.17		0.95		−18.8 *				
		T3: 13	1.15		0.98		−14.8 *				
**Tordi 2001 [43]**	**W**	5		17		25.4		+49.4 *	32	72	+125 *
**Van den Berg 2010 [44]**	**W**	9	1.16		0.95		−18.1 *				
**Abonie 2021 [46]**	**H**	9	0.96		0.80		−16.7 *				

A, arm crank ergometer; W, wheelchair; H, handcycle; T1–3, training group 1–3; VO_2_, oxygen uptake; PO, power output; submax, submaximal; *, ** significance of the difference between pre-training and post-training at the level of *p* < 0.05, 0.01, respectively.

**Table 6 biology-12-00355-t006:** Transfer effects on VO_2peak_ and PO_peak_ from arm training to legs.

Study (Author, Year)	Test Device	n	VO_2peak ARM_						VO_2peak LEG_						PO_peak ARM_			PO_peak LEG_		
			Pre-Test		Post-Test		% Change		Pre-Test		Post-Test		% Change		Pre-Test (w)	Post-Test (W)	% Change	Pre-Test (W)	Post-Test (W)	% Change
			(L × min^−1^)	(ml × min^−1^ × kg^−1^)	(L × min^−1^)	(ml × min^−1^ × kg^−1^)	(L × min^−1^)	(ml × min^−1^ × kg^−1^)	(L × min^−1^)	(ml × min^−1^ × kg^−1^)	(L × min^−1^)	(ml × min^−1^ × kg^−1^)	(L × min^−1^)	(ml × min^−1^ × kg^−1^)						
**Bhambhani** **1991 [26]**	A/CYC	8	2.72	32.0	3.15	37.5	+15.8 *	+17.2 *	3.77	44.5	3.84	45.7	+1.9	+2.7	96.4	108.7	+12.8 *	250	264.7	+5.9
**Hill 2018 [28]**	A/CYC	10	1.12	17	1.39	22	+24.1 *	+29.4 *	1.44	23	1.64	26	**+13.8 ***	**+13 ***	51	65	+27.5 *	98	108	**+10.2 ***
**Lewis 1980 [24]**	A/CYC	5	1.64		2.22		+35.4 *		2.69		3.02		+12.3							
**Loftin 1988 [30]**	A/CYC	19					+33 *	+32 *					**+7 ***	**+7 ***						
**Magel 1978 [31]**	A/TM	9	2.69	33.9	3.13	39.3	+16.4 *	+15.9 *	4.48	56.4	4.57	57.2	+2	+1.4						
**Pogliaghi 2006 [33]**	A/CYC	6	1.62	22.0	1.99	26.8	+22.8 *	+21.8 *	2.31	31.3	2.52	33.8	**+9.1 ***	**+8 ***	87	106	+21.8 *	158	170	**+7.6 ***
**Stamford 1978 [37]**	A/CYC	9	2.82	36.9	3.3	44	+17 *	+19.2 *	3.2	42.7	3.2	43.1	+/-0	+0.9						
**Tordi 2001 [43]**	W/CYC	5		30.4		39.3	+29.3 **			46		47		+2.2	66	108	+63.6 ***	228	228	+/-0
**Hettinga 2016 [47]**	H/CYC	11	1.9	28.3	2.2	33.2	+18.1 *	+17.3 *	3.2	47.1	3.1	46.7	-1.1	-0.8	89	117.4	+31.9 *	274.5	278.2	+1.3

A, arm crank ergometer; W, wheelchair; H, handcycle; CYC, cycle; TM, treadmill; T1–3, training group 1–3; VO_2_, oxygen uptake; PO, power output; *, **, *** significance of the difference between pre-training and post-training at the level of *p* < 0.05, 0.01, 0.001, respectively.

**Table 7 biology-12-00355-t007:** Transfer effects on VO_submax_ and PO_submax_ from arm training to legs.

Study (Author, Year)	Test Device	n	VO_submax ARM_						VO_2submax LEG_						PO_submax ARM_			PO_submax LEG_		
			PRE-TEST		Post-Test		% Change		Pre-Test		Post-Test		% Change		Pre-Test (w)	Post-Test (W)	% Change	Pre-Test (W)	Post-Test (W)	% Change
			(L × min^−1^)	(ml × min^−1 x^kg^−1^)	(L × min^−1^)	(ml × min^−1^ × kg^−1^)	(L × min^−1^)	(ml × min^−1^ × kg^−1^)	(L × min^−1^)	(ml × min^−1^ × kg^−1^)	(L × min^−1^)	(ml × min^−1^ × kg^−1^)	(L × min^−1^)	(ml × min^−1^ × kg^−1^)						
**Bhambhani 1991 [26]**	A/CYC	8	1.18	13.8	1.46	17.4	+23.7 *	+26.1 *	2.30	27.5	2.31	28.0	+0.43	+1.81	50.7	59.8	+17.9 *	143.3	158.0	+10.5
**Clausen 1973 [16]**	A/CYC	5	1.25		1.15		−8		1.64		1.63		−0.6							
**Klausen 1974 [17]**	A/CYC	5	1.25		1.15		−8		1.64		1.63		−0.6							
**Lewis 1980 [24]**	A/CYC	5	1.24		1.08		−12.9 *		1.37		1.28		−**6.6 ***							
**McKenzie 1978 [18]**	A/CYC	7	1.36		1.12		−17.6 *		1.64		1.50		−**8.5 ***							
**Pogliaghi 2006 [33]**	A/CYC	6	1.07		1.26		+17.8 *		1.65		1.74		**+5.5 ***		60	70	+16.7 *	110	115	**+9.1 ***
**Rasmussen 1975 [34]**	A/CYC	5	1.76		1.64		−6.8		2.61		2.64		+1.1							
**Tordi 2001 [43]**	W/CYC	5		17		25.4		+49.4 *		28		34	**+21.4 ***		32	72	+125 *	132	156	**+18.2 ***

A, arm crank ergometer; W, wheelchair; CYC, cycle; T1–3, training group 1–3; VO_2_, oxygen uptake; PO, power output; submax, submaximal; * significance of the difference between pre-training and post-training at the level of *p* < 0.05.

**Table 8 biology-12-00355-t008:** Overview: Outcomes of UBET.

Study (Author, Year)	Test Device	n	Training Design		Increase/Decrease in VO_2peak ARM_	Increase/Decrease in VO_2submax ARM_	Increase/Decrease in VO_2peak LEG_	Increase/Decrease in VO_2submax LEG_
			Intensity (High/Moderate/Low)	Length (Days/Weeks)				
**Bhambhani** **1991 [26]**	A	8	Moderate	3 d × wk^−1^ 8 wk	**+**	**+**	+ (not significant)	+ (not significant)
**Clausen 1973 [16]**	A	5	High	5 d × wk^−1^ 5 wk	not measured	- (not significant)	- (not significant)	- (not significant)
**Klausen 1974 [17]**	A	5	High	5 d × wk^−1^ 5 wk	not measured	- (not significant)	- (not significant)	- (not significant)
**El-Sayed 2004 [27]**	A	7	Moderate	3 d × wk^−1^ 12 wk	**+**	not measured	not measured	not measured
**Hill 2018 [28]**	A	10	Moderate	3 d × wk^−1^ 6 wk	**+**	not measured	**+**	not measured
**La Monica 2019 [29]**	A	T1: 11	High	3 d × wk^−1^ 2 wk	+ (not significant)	not measured	not measured	not measured
		T2: 11	High		+ (not significant	not measured	not measured	not measured
		T3: 10	High		**+**	not measured	not measured	not measured
**Lewis 1980 [24]**	A	5	Moderate	4 d × wk^−1^ 11 wk	**+**	**-**	+ (not significant)	**-**
**Loftin 1988 [30]**	A	19	Moderate/ High	4 d × wk^−1^ 5 wk	**+**	not measured	**+**	not measured
**Magel 1978 [31]**	A	9	High	3 d × wk^−1^ 10 wk	**+**	- (not significant)	+ (not significant)	not measured
**McKenzie 1978 [18]**	A	7	High	5 d × wk^−1^ 5 wk	not measured	**-**	not measured	**-**
**Pinto 2019 [32]**	A	T1: 10	High	2 d × wk^−1^ 6 wk	+ (not significant)	not measured	not measured	not measured
		T2: 10	High		**+**	not measured	not measured	not measured
**Pogliaghi 2006 [33]**	A	6	Moderate	3 d × wk^−1^ 12 wk	**+**	**+**	**+**	**+**
**Rasmussen 1975 [34]**	A	5	Moderate/High	5 d × wk^−1^ 5 wk	not measured	- (not significant)		+ (not significant)
**Sedlock 1988 [35]**	A	6	High	3 d × wk^−1^ 5 wk	+ (not significant)	+ (not significant)	not measured	not measured
**Simmons 1978 [36]**	A	10	High	2 d × wk^−1^ 4 wk	**+**	*Graphic/no exact data	*Graphic/ no exact data	*Graphic/ no exact data
**Stamford 1978 [37]**	A	8	High	3d × wk^−1^ 10wk	**+**	not measured	did not change	not measured
**De Groot 2008 [38]**	W	14	Low	3 d × wk^−1^ 7 wk	**+**	**-**	not measured	not measured
**De Groot 2013 [39]**	W	T1: 14	Low	3 d × wk^−1^ 7 wk	**+**	**-**	not measured	not measured
		T2: 10	Low		- (not significant)	**-**	not measured	not measured
		T3: 13	Moderate		**+**	**-**	not measured	not measured
**Glaser 1981 [40]**	W	7	High	3 d × wk^−1^ 7 wk	*Graphic/no exact data	*Graphic/no exact data	*Graphic/no exact data	*Graphic/ no exact data
**Goosey-Tolfrey 2011 [41]**	W	T1: 8	Moderate	3 d × wk^−1^ 3 wk	**+**	not measured	not measured	not measured
		T2: 8	Moderate		**+**	not measured	not measured	not measured
**Grange 2002 [42]**	W	7	Moderate/High	3 d × wk^−1^ 6 wk	**+**	not measured	not measured	not measured
**Tordi 2001 [43]**	W	5	Moderate/High	3 d × wk^−1^ 6 wk	**+**	+	+ (not significant)	+
**Van den Berg 2010 [44]**	W	10	Low	3 d × wk^−1^ 7 wk	- (not significant)	**-**	not measured	not measured
**Van der Woude 1999 [45]**	W	T1: 9	Low	3 d × wk^−1^ 7 wk	+ (not significant)	not measured	not measured	not measured
		T2: 10	Moderate		**+**	not measured	not measured	not measured
**Abonie 2021 [46]**	A	9	Low	3 d × wk^−1^ 7 wk	+ (not significant)	**-**	not measured	not measured
**Hettinga 2016 [47]**	A	11	Moderate	3 d × wk^−1^ 7 wk	**+**	not measured	- (not significant)	not measured
**Schoenmakers 2016 [48]**	A	T1: 8	Moderate	3 d × wk^−1^ 7 wk	**+**	not measured	not measured	not measured
		T2: 8	High		**+**	not measured	not measured	not measured

A, arm crank ergometer; W, wheelchair; H, handcycle; CYC, cycle; TM, treadmill; T1–3, training group 1–3; VO_2_, oxygen uptake; PO, power output; submax, submaximal; low, ≅ 30–50% HRR, HR_peak_, VO_2peak_; moderate, ≅ 60–80% HRR, HR_peak_, VO_2peak_; high, ≅ >80% HRR, HR_peak_, VO_2peak._

## Data Availability

No new data were created or analyzed in this study.

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
