# Peer review of "Effects of Upper Body Exercise Training on Aerobic Fitness and Performance in Healthy People: A Systematic Review"

_biology, 2023, doi:10.3390/biology12030355_

Round 1

Reviewer 1 Report

The present work is an important addition to numerous studies on determining the training effect by lower extremity (leg) activity.Training of upper extremity trainingThis has a great importance for people with disabilities such as paraplegia or peripheral vascular diseases.
The literature search and meta-analysis have been done correctly and the conclusions are clear.
The discussion should consider one more study by  Saltin et al., these authors published a study with one-leg training using pedal crank ergometry,where only the trained leg  showed training success. This work should be discussed, it gives a further explanation of the present findings.
Thereafter this paper can be be accepted.

Reviewer 2 Report

Manuscript review: Effects of upper body exercises on VO2 in healthy people: A systematic review.

Thank you very much for the opportunity to review your manuscript.

At the outset, I would like to point out that it is methodologically correct. However, it contains too many generalities

1 The manuscript is written on a 2021 template. The authors have made no effort to correct this.

2. The literature review ends with 2021, no insight into manuscripts from 2022 in the topic under study.

3. Responses to exercise or training of healthy subjects and those with paraplegia or tetraplegia cannot be combined together. There are different adaptive mechanisms in the response to exercise.

4. Upper limb exertion is not indicated not only in cardiac rehabilitation but also in pulmonary rehabilitation. This was not mentioned by the authors.

5. No reference is made to differences in muscle metabolism and hormonal and neurohormonal response to lower and upper limb efforts.

(6) Based on the comments cited, I recommend that the manuscript be submitted, after correction of the text, to Healthcare, Journal of Functional Morphology and Kinesiology, Sports or other sports journals.

Reviewer 3 Report

This is an important contribution to the exercise sciences, particularly as arm/upper body exercise provides additional opportunities for healthy adults to reach fitness goals especially when the lower extremities are injured.

However, a careful review and re-write is necessary to clear up some manuscript organizational, grammatical and syntax problems throughout the paper.

I would like to see a more fulsome and potent expression of the WHY this research meta analysis is important and is expressed clearly in the introduction of the manuscript - the current text does not decisively express this aim.

This could be improved by incorporating Section 1.1 into the introduction. 
One confusing aspect is the inclusion/exclusion of spinal cord injured persons in Section 1.1 and in several points in the paper, when the aim was to exclude them in the analysis (i.e., Table 3): Please resolve.

Some portions of the manuscript can be streamlined and cleaned up: mainly by eliminating some tables, and incorporating data from other tables into others.

Section 1.2 is a full description of the physiology of UBET: I believe that a brief survey of the differences between UB and lower body endurance training is necessary (cardiovascular and metabolic differences), but not in the detail here - carry this through in the discussion section.  There is an inappropriate discussion citation of the author's own work in this area in the early portion of the paper (line 102)- probably belongs in the main data set, results and discussion, vs. the introduction part of the paper.

Specific Suggestions:

Abstract: in relation to transfer effects, in Abstract they are described as "indecisive" (line 12/13), whereas later in the Abstract, they are characterized as "can be observed" (Line 35) . Please resolve.

Section 2.0  Table 1 is not necessary - very cluttered and data bases collection details are described in the text.

Table 2. The title needs re-wording: the table is not a summary of the 'effect' or 'performance' of UBET, but rather summarizes the studies' description, subject recruitment, design and measures. 

Table 3.  Same as above: please re-title.

Table 4. Same as above: please re-title

Section 3.4: "Training Effects" this nested section heading does not contain any text.

Heading 3.4.1: Suggest re-title to "Maximal UBET Responses"

Heading 3.4.2: same: suggest re-title the heading to read "Submaximal UBET Responses"

Section 3.4.3  Transfer Effects: This paragraph is somewhat confusing and difficult to track:  Line 364, it states that all observed transfer effects..", but later, 3 of 9 studies (maximal responses), showed transfer effects (line 371). Two studies demonstrated increase in VO2Peak leg (line 371). Please re-write this section to more clearly outline the results (some are non-significant, some are, depending upon the measures observed (max vs. submax etc).  Also, please cite the three key studies demonstrating transfer effects by Author's in the text (so we can cross reference Tables 7 and 8.

Tables 7 and 8: please clearly denote the significance levels assigned to the key studies (star symbol in the table, and symbol definition and P level in the captions)

Table 9: could these data be incorporated into the other tables?

Section Headings 4.1, 4.11, 4.111:  please correct and clarify heading nesting..

Discussion:  quite lengthy and concepts are discussed in difficult detail. Suggest some re-organization to improve flow for the readership. Here is where the discussion of cardiovascular differences between UB and LB (hypertension effect, venous return, RPP, etc.) found in the introduction, can be inserted.  

The discussion of transfer effects should be strengthened - with a clear analysis of the results and why only 3 of the 20-odd studies failed to demonstrate transfer effects: Some possible factors are discussed but not in a systematic way: here is a good opportunities to discuss these by splitting these giant paragraphs into smaller packages by using sub headings. For example, sub-heading titles like: study design, training methods, training devices, training type (fixed, continuous, interval, above VT, lactate Threshold, etc.,), subject age, hetrogeneity/homogeneity, gender, co-morbidities, etc.  

Section 4.2: the purpose of this paragraph is questionable: suggest combining into 4.3 Conclusions (and adding recommendations for future research there).

Section 5 Conclusions: The bullet points summarizing the main findings are good, but with the exception of the last one pertaining to cardiac patients: since this population was not collected/discussed in the paper, why raise this population at this point? suggest omitting.  (as an aside, however, this point might highly contestable within the cardiac rehabilitation literature, as many studies have found upper body exercise training to be safe and effective in these patients, when prescribed correctly). 

Round 2

Reviewer 2 Report

The authors made a great deal of corrections to their manuscript. They have included many new variables. The layout has also been improved. 

please correct the position of literature 23, 19 

the notation VO2max should be as in the manuscript in subscript

Author Response

Reviewer #2:

Thank you again very much for your helpful and positive feedback. We incorporated your comments into the manuscript (see details below).

Comments and Suggestions of Reviewer 2:

The authors made a great deal of corrections to their manuscript. They have included many new variables. The layout has also been improved. 

Point 1: please correct the position of literature 23, 19 

Reply: We have checked references 19 and 23 in the text as well as in the source list and we cannot find any incorrect assignment. Or we misunderstand your comment. If so, could you make that comment more specific?

Point 2: the notation VO2max should be as in the manuscript in subscript

Reply: Thank you for this comment. In reference 19, we have changed the notation VO2max to subscript form (VO2max).

Reviewer 3 Report

The revision has resulted in a much improved manuscript; the organization of the paper has resulted in a much better flow of information and improved readability. Omitting of redundant materials, tables and text has improved the manuscript.  
There are remaining a few minor spelling/syntax issues, which may be delt with at the editorial/clerical stage of manuscript production

Author Response

Reviewer #3:

Thank you very much for your positive feedback.

Comments and Suggestions of Reviewer 3:

The revision has resulted in a much improved manuscript; the organization of the paper has resulted in a much better flow of information and improved readability. Omitting of redundant materials, tables and text has improved the manuscript.  

Point 1: There are remaining a few minor spelling/syntax issues, which may be delt with at the editorial/clerical stage of manuscript production.

Reply: Thank you for this comment. Another native speaker corrected the text and improved final mistakes.